# Identification of a Conserved, Linear Epitope on VP3 of *Enterovirus A* Species Recognized by a Broad-Spectrum Monoclonal Antibody

**DOI:** 10.3390/v15041028

**Published:** 2023-04-21

**Authors:** Lie Fu, Xiao-Yu Zhang, Wei-Ping Jin, Chen Wang, Sha-Sha Qian, Meng-Jun Wang, Wen-Hui Wang, Sheng-Li Meng, Jing Guo, Ze-Jun Wang, Xiao-Qi Chen, Shuo Shen

**Affiliations:** Wuhan Institute of Biological Products Co., Ltd., Wuhan 430207, China; fl932356@163.com (L.F.);

**Keywords:** broad-spectrum monoclonal antibody, *Enterovirus A* species, linear and conserved epitope

## Abstract

Outbreaks of hand, foot and mouth disease (HFMD) have occurred frequently in the Asian-Pacific region over the last two decades, caused mainly by the serotypes in *Enterovirus A* species. High-quality monoclonal antibodies (mAbs) are needed to improve the accuracy and efficiency of the diagnosis of enteroviruses associated HFMD. In this study, a mAb 1A11 was generated using full particles of CV-A5 as an immunogen. In indirect immunofluorescence and Western blotting assays, 1A11 bound to the viral proteins of CV-A2, CV-A4, CV-A5, CV-A6, CV-A10, CV-A16, and EV-A71 of the *Enterovirus A* and targeted VP3. It has no cross-reactivity to strains of *Enterovirus B* and *C*. By mapping with over-lapped and truncated peptides, a minimal and linear epitope _23_PILPGF_28_ was identified, located at the N-terminus of the VP3. A BLAST sequence search of the epitope in the NCBI genus *Enterovirus* (taxid: 12059) protein database indicates that the epitope sequence is highly conserved among the *Enterovirus A* species, but not among the other enterovirus species, first reported by us. By mutagenesis analysis, critical residues for 1A11 binding were identified for most serotypes of *Enterovirus A*. It may be useful for the development of a cost-effective and pan-*Enterovirus A* antigen detection for surveillance, early diagnosis and differentiation of infections caused by the *Enterovirus A* species.

## 1. Introduction

Hand, foot and mouth disease (HFMD) is an epidemic disease frequently occurring in infants and young children. More than 90% of the reported cases were caused by serotypes of the *Enterovirus A* species within the genus *Enterovirus* of the family *Picornaviridae* in China [1,2,3,4]. The genus *Enterovirus* includes nine enterovirus species and three rhinovirus species. The nine enterovirus species contain *Enterovirus A*, *B*, *C*, *D*, *E*, *F*, *G*, *H*, and *J*. The *Enterovirus A* includes multiple serotypes on the basis of antibody neutralization tests and sequence homology, including coxsackieviruses A2 (CV-A2), CV-A3, CV-A4, CV-A5, CV-A6, CV-A7, CV-A8, CV-A10, CV-A12, CV-A14, CV-A16; enteroviruses A71 (EV-A71), EV-A76, EV-A89, EV-A90, EV-A91, EV-A92, EV-A114, EV-A119, EV-A120, EV-A121; simian enteroviruses SV19, SV43, SV46; and baboon enterovirus A13 (BA13) [5,6,7,8,9,10]. Meanwhile, large numbers of enterovirus serotypes can be divided into four human enterovirus (HEV) species, designated HEV-A to D and variants within a serotype are further divided into subgenotypes, clades or lineages. The HEV-A species caused significantly more HFMD than HEV-B species [5,6]. Recent epidemiological studies have found that CV-A6 was the predominant serotype causing HFMD, followed by CV-A16, CV-A10, CV-A5, CV-A2, CV-A4 and EV-A71 in mainland China since the monovalent EV-A71 vaccine was available [1].

The enterovirus genome is a single stranded RNA molecule of approximately 7400 nucleotides, consisting of a single open reading frame (ORF) flanked by 5′-untranslated regions (5′-UTR) and 3′-untranslated regions (3′-UTR). The ORF is translated into a polyprotein which is co- and post-translationally cleaved into four structural proteins VP4, VP2, VP3, and VP1 by virus-encoded proteases, which form the viral capsid, and nonstructural proteins which are mainly involved in the processing of the polyprotein and replication of the virus. The VP1 to VP3 are arranged in a wedge-shaped, eight-stranded antiparallel β-barrel linked by loops, which present exposed epitopes for receptor and antibody binding [11,12]. The naked capsid is an icosahedral shell with pseudo-T = 3 symmetry, approximately 30 nm in diameter. The conformations of virus particles change dynamically and basically switch between a native and an expanded form. The major particles are full particles and empty particles. Full particles contain viral RNA and are infectious following maturation cleavage of VP0 to VP4 and VP2. Empty particles are mostly expanded in diameter without viral RNA and with un-cleaved VP0. VP1 to VP3 are mostly exposed on the virion surface and their N-termini are located inside the capsid shell, while VP4 is completely internal to the virion [13,14,15,16]. The serotype of an enterovirus is defined by neutralizing epitopes generally located on surface-exposed loops of the virus capsid proteins, particularly on VP1 but also on VP2 and VP3 [17,18].

The symptoms of HFMD are usually mild, comprising fever, loss of appetite, sore throat, and a rash with blisters, which do not need specific treatment. Atypical skin findings in HFMD may be seen in children with atopic dermatitis. These include ‘eczema coxsackium’, in which eczematous skin is superinfected with coxsackievirus, resembling a herpes infection. Nail changes, such as shedding, may follow HFMD after a latency period. However, there are uncommon neurological or cardiac complications, such as meningitis and acute flaccid paralysis, that can be fatal [19,20,21,22,23].

At present, the diagnosis of HFMD is mainly based on the detection of viral RNA by reverse transcription polymerase chain reaction (RT-PCR), viral specific IgM by enzyme-linked immunosorbent assay (ELISA) or an IgM-colloidal gold immunochromatographic assay (GICA). RT-PCR is superior in rapidity and sensitivity to virus isolation and is recommended as the primary diagnostic tool for enteroviruses. In developing countries such as Malaysia, the laboratory capacity to carry out EV-A71 IgM detection is greater than that of the gold-standard methods of virus culture or molecular detection. The sensitivity, specificity, positive predictive value, and negative predictive value rates were similar between ELISA and GICA [24,25,26,27,28,29,30]. Meanwhile, the pan-HEV mAb 5-D8/1 and a mixture of several mAbs have been used in laboratory diagnosis and identification of *Enterovirus A*, *B* and *C*, but species-specific mAbs have not been generated [31,32].

Monoclonal antibodies are efficient analytic tools for the detection, screening and characterization of biomolecules, and have a wide range of applications as diagnostics and immunotherapeutics. Most of the reported monoclonal antibodies are serotype-specific monoclonal antibodies, which have some limitations in diagnosing multiple serotypes simultaneously [33,34,35,36,37]. Aiming to develop highly specific and reactive mAbs for the detection of the enteroviruses associated with HFMD, a broad-spectrum mAb (1A11) was produced, which has not been reported previously. The mAb 1A11 was evaluated for its application in various immunoassays and might be useful for laboratory diagnosis of HFMD caused by the infection of *Enterovirus A*.

## 2. Materials and Methods

### 2.1. Ethics Statement

Animal experiments were performed in accordance with the guidelines of the Standardization Administration of China [38]. Experimental protocols were approved by the Animal Ethics Committee of the Wuhan Institute of Biological Products (WIBP-A II 382020003).

### 2.2. Cells, Viruses and Anti-Serum

Human rhabdomyosarcoma cells (RD) and African green monkey kidney cells (Vero) were obtained from the American Type Culture Collection (ATCC). The cells were grown in Dulbecco’s Modified Eagle Medium (DMEM, Gibco, New York, NY, USA), supplemented with 10% fetal bovine serum (FBS, Tianhang, Hangzhou, China), 90 µg of streptomycin per ml, and 90 units of benzylpenicillin per ml (Sinopharm Chemical Reagent, Shanghai, China) at 37 °C in a humidified 5% CO_2_ incubator. Strains of CV-A5, CV-A2, CV-A4, CV-A6, CV-A10, CV-A16, EV-A71 and echovirus 11 were isolated from HFMD patients in Xiang Yang, China, in 2016 and 2017 in RD or Vero cells [1]. Sabin 3 of poliovirus and rotavirus G8 (Rota-G8) were vaccine strains kept at the Wuhan Institute of Biological Products. The empty particles (EP) and full particles (FP) of different strains were purified by sucrose cushion, sucrose gradient and CsCl density gradient ultracentrifugation sequentially, as previously described [39]. Purified viruses were obtained by mixing the empty and full particles in the original proportions and used in the assays. Rabbit anti-CV-A5 serum was prepared as described previously [39]. Purified rotavirus G8 was provided by the Rotavirus vaccine laboratory of the Wuhan Institute of Biological Products.

### 2.3. Generation of Monoclonal Antibody (mAb)

BALB/c mice of 6 to 8-weeks old were immunized subcutaneously with 60 μg of purified FP of CV-A5 in 0.1 mL of phosphate-buffered saline (PBS), which was emulsified with an equal volume of Freund’s complete adjuvant (Sigma, St. Louis, MO, USA). Mice were boosted every 2 weeks with the same doses of antigen in Freund’s incomplete adjuvant for three times. Thereafter, mice were boosted intraperitoneally with 60 μg of the same antigen 3 days before the fusion of splenocytes with SP2/0 cells. Hybridomas were selected with hypoxanthine-aminopterin-thymidine (HAT) medium (Sigma, St. Louis, MO, USA) and supernatants were screened by ELISA as described below. Isotypes of the selected mAbs were determined using a mouse antibody isotyping kit (Sino Biological, Peking, China).

### 2.4. Indirect Immunofluorescence Assay (IFA)

RD cells were seeded in 6-well cell culture plates. The cells were infected with 7 strains of CV-A2, CV-A4, CV-A5, CV-A6, CV-A10, CV-A16, and EV-A71 of *Enterovirus A*, 1 strain of Echovirus 11 of *Enterovirus B*, and 1 strain of poliovirus Sabin 3 of *Enterovirus C* for 24–48 h. The infected cell monolayers were fixed with 4% paraformaldehyde for 30 min at room temperature, followed by permeabilization with 0.1% Triton-X 100 in PBS for 10 min at room temperature. Cells were then washed with PBS and blocked with PBS containing 1% bovine serum albumin (BSA) at 37 °C for 30 min before being incubated with 1 μg /mL mAb 1A11 for 1 h at 37 °C. Cells were rinsed three times with PBS and incubated with 0.2 μg/mL of FITC-labeled goat anti-mouse IgG (Thermo Fisher, Waltham, MA, USA) for 1 h at 37 °C. The cells were rinsed again in PBS and examined under an inverted fluorescence microscope (Leica, Wetzlar, Germany).

### 2.5. Western Blotting

The specificity and reactivity of the mAb 1A11 were also determined by Western blotting using purified viruses including CV-A2, CV-A4, CV-A5, CV-A6, CV-A10, CV-A16, EV-A71, Echo11, Sabin 3, Rota-G8 and Vero cell lysate. Protein samples were denatured in the SDS-PAGE loading buffer (containing 5% β-mercaptoethanol) at 100 °C for 10 min. Equal amounts of viral proteins were separated on a 4–20% SDS-PAGE gel (Genscript, Nanjing, China) at 100 volts for 2 h, and then transferred to nitrocellulose membranes (NC membranes) by TransBlot (BioRad, Hercules, CA, USA). Nitrocellulose membranes were blocked with PBS containing 1% bovine serum albumin (BSA) and incubated with rabbit anti-CV-A5 serum at a dilution of 1:5000, or with mAb 1A11 at 0.2 μg/mL for 1 h each. Membranes were washed three times for 15 min each in PBST (PBS containing 0.1% tween-20) before incubating in 0.1 μg/mL of Secondary horseradish peroxidase (HRP)-conjugated goat anti-mouse IgG (Boster, Wuhan, China) for 1 h. Membranes were washed again three times for 15 min each in PBST followed by incubation with a Western chemiluminescent HRP substrate (Millipore, Temecula, CA, USA) and were detected by an Amersham Image Quant 800 (Cytiva, Marlborough, MA, USA).

### 2.6. Indirect ELISA

To analyze the affinity of mAb 1A11 against different strains, 96 wells of ELISA plates were coated with purified viruses including CV-A2, CV-A4, CV-A5, CV-A6, CV-A10, CV-A16, EV-A71, Echo11, Sabin 3, and Rota-G8, at 1 μg/mL, and incubated overnight at 4 °C in carbonate buffer solution (pH 9.6). The plates were washed and blocked with 1% BSA in PBST for 1 h at 37 °C. The serial dilutions of mAb 1A11 were added into appropriate wells and incubated at 37 °C for 1 h. The wells were rinsed four times with PBST and incubated with HRP-conjugated goat anti-mouse IgG at 37 °C for 1 h. The wells were incubated in 100 μL of tetramethyl benzidine (TMB, Sigma, St. Louis, MO, USA) for 15 min at 37 °C and the reaction was then stopped with 2 M H_2_SO_4_. The absorbance was measured at 450 nm with a microplate reader (Thermo Fisher, Waltham, MA, USA).

For the detection of the linear epitopes of the mAb 1A11, the 96 wells plates were coated with synthetic peptides at 2 μg/mL, and the concentration of the detection antibody 1A11 was 1 μg/mL, and the other steps were the same as described above.

### 2.7. Epitope Mapping of mAb

The mAb was confirmed to bind to the CV-A5 VP3 protein by Western blotting analysis. A series of 31 peptides spanning the full-length VP3 region of CV-A5 were synthesized by Genscript (Nanjing, China). Each peptide consists of about 20 amino acid residues and has 10 residues that overlap with the adjacent peptides. The 31 overlapping peptides were used to detect the binding region of the mAb by ELISA. Then truncated peptides of the binding region were synthesized to determine the minimum epitope needed for binding.

### 2.8. BLAST of mAb Epitope

The mAb 1A11 defined epitope sequences and flanking sequences of CV-A5 were aligned with those of selected strains in this study by SnapGene. The amino acid sequence of the antigen epitope was BLAST analysis with the NCBI genus *Enterovirus* protein database (taxid: 12059) to find the sequences with one amino acid mutations. These mutant sequences were synthesized for ELISA to further confirm the antigen epitope spectrum of the mAb.

## 3. Results

### 3.1. Generation and Characterization of mAb 1A11

Purified CV-A5 was used to inoculate BALB/c mice and to generate the mAb 1A11 using hybridoma technology as described previously [40]. The subclass of 1A11 was isotyped to be IgG1. Examined by a microneutralization assay, 1A11 did not demonstrate any neutralization activity to CV-A5. Western blotting revealed that 1A11 bound to the 26 kDa protein of the purified empty and full particles of CV-A5, which corresponded to the predicted VP3 in size, using rabbit anti-CV-A5 serum and lysate of uninfected Vero cells as negative controls (Figure 1). The results showed that 1A11 reacted with a denatured, linear epitope on the capsid protein VP3.

### 3.2. Binding Specificity of the Antibody to the Enterovirus A Species

Experiments were conducted to determine whether the antibody could bind to the VP3 of other serotypes of *Enterovirus A* and representative serotypes of other Human enterovirus species that were stored in our labs. Seven available serotypes, CV-A2, CV-A4, CV-A5, CV-A6, CV-A10, CV-A16, and EV-A71 of *Enterovirus A*, echovirus 11 of *Enterovirus B*, and Sabin 3 of poliovirus of *Enterovirus C* were used in the IFA and WB assays. As shown in Figure 2a, the mAb 1A11 was able to stain the cells infected by all seven serotypes of *Enterovirus A* strongly, but could not stain cells infected by echovirus 11, poliovirus Sabin 3 and a mock-infected in IFA. The binding specificity was also confirmed by Western blotting (Figure 2b). The mAb 1A11 bound to the VP3 of the seven serotypes of *Enterovirus A*, but not to the VP3 of echovirus 11 and poliovirus Sabin 3 of *Enterovirus B* and *C*. There were no cross-reactivities to lysates of cells infected by rotavirus G8 and mock-infected.

The specificity and affinity of mAb 1A11 against *Enterovirus A* was further confirmed by ELISA with these nine serotypes of *Enterovirus A*, *B* and *C* and a non-enterovirus control rotavirus G8. The mAb 1A11 showed activity in its binding to all seven serotypes of *Enterovirus A* but not to the serotypes of *Enterovirus B* and *C,* and rotavirus G8. At a mAb concentration of 10 ng/mL, 1A11 had similar high affinity against six serotypes of CV-A2, CV-A4, CV-A5, CV-A6, CV-A10, and CV-A16, and slightly less affinity to EV-A71, and no affinity to Echo11, Sabin3, or rotavirus G8 (Figure 3). This result verified the recognition specificity of 1A11 against the serotypes of *Enterovirus A*.

### 3.3. Epitope Mapping of mAb 1A11

To define the linear epitopes, the 31 peptides, each of about a 20-mer with 10 overlapping residues at the ends, that span the entire VP3 region, were synthesized for reactivity assays with the 1A11. Peptide ELISAs showed that the 1A11 reacted strongly with the peptide 3 (_16_TDDEVSAPILPGFQPTPEI_34_) but could not react to other peptides (Figure 4a). Amino acid sequence alignment of the sequences of representative serotypes of the peptide 3 region predicted a highly conserved epitope _21_SAPILPGF_28_ in *Enterovirus A* (Figure 4b). To further map the binding epitopes, ELISAs using truncated VP3 peptides from both ends of a core sequence were performed. As shown in Figure 4c, 1A11 could react with truncated peptides a-c and f, but not with d, e and g-j. The results indicated that deletion up to P_23_ from the N-terminus or F_28_ from the C-terminus of _21_SAPILPGF_28_ eliminated the binding of the truncated peptides by mAb 1A11. Therefore, it was confirmed that the motif _23_PILPGF_28_ is the defined minimal epitope, and P_23_ or F_28_ is critical for 1A11 binding.

### 3.4. BLAST Analysis of 1A11 Epitope

BLAST analysis of this epitope against the NCBI *Enterovirus A* protein database (taxid: 138948) belonging to genus *Enterovirus* protein database (taxid: 12059) indicates that this epitope is relatively conserved in most of published *Enterovirus A* VP3 sequences (3654 sequences in total), except for 1498 sequences. These 1498 sequences fall into seven categories of variations, each of which has one or two amino acid mutations within the epitope (Table 1). In addition, BLAST analysis of this epitope against the other enterovirus species (*Enterovirus B* to *J*) protein databases found only three variants with one amino mutation present in *Enterovirus D* (Table 1). To study the effects of these substitutions on the binding ability of the mAb 1A11, nine peptides were synthesized and the binding abilities were tested by ELISA. The mutations of PILPNF (G to N at position twenty-seven) in fourteen-thousand eighty-seven EV-A71 strains and one CV-A16 strain, PVLPGF (I to V at position twenty-four) in one EV-A125 strain and PILPCF (G to C at position twenty-seven) in three EV-D68 strains had no influence on their reactivity to 1A11 (Table 1 and Figure 5). These three EV-D68s might be recombinants between *Enterovirus A* and *D*. The results indicate that single mutations of these residues at these positions are tolerable. On the contrary, the other few mutations, P to H at position 26, G to R at position 27 and F to Y at position 28 found in CV-A10, EV-A91, EV-A121 and SV19, respectively, were not recognized by 1A11 (Table 1 and Figure 5). Combined with mutagenesis analysis, more than 99% of *Enterovirus A* and rare *Enterovirus D* variants listed in the NCBI enterovirus species protein database might be recognized by mAb 1A11.

## 4. Discussion

At present, the *Enterovirus A* serotypes of CV-A2, CV-A4, CV-A5, CV-A6, CV-A10, CV-A16 and EV-A71, have caused more than 90% of the cases of HFMD in China [1]. Previous studies on non-neutralizing mAbs against HFMD-associated *Enterovirus A* mainly concentrated on serotype-specific or strain-specific mAbs within a single serotype. The monoclonal antibody 7C7, for example, was mapped to amino acids _142_EDSHP_146_ of the VP2 capsid protein of EV-A71 and did not cross-react to CV-A4, CV-A6, CV-A10 or CV-A16 [37]. The mAb 1D9 detected successfully the linear epitope RVADVI on the VP1 protein of EV-A71 and showed no cross-reactivity to the four selected strains of CV-A4, CV-A6, CV-A10, and CV-A16 [33]. In an earlier study, a group-specific mAb 5-D8/1 was generated which cross-reacted with the VP1 of Coxsackieviruses A and B, and echoviruses and polioviruses. This antibody proved particularly useful in the detection of enterovirus antigens in identifying field isolates of the group of viruses and in circulating immune complexes. It was also shown to be practical and informative in immunohistochemistry of enterovirus diagnosis and research [31,41]. A broad-spectrum antibody 3A6, recognizes VP1 of several EV-Bs and also the EV-C representative poliovirus 3. It performed well in multiple immunological experiments including ELISA, immunoelectron microscopy, immunocyto- and histochemistry and in western blotting, detecting EVs in infected cells and tissues [42]. Miao and colleagues reported the combination of pan-*Human enterovirus A* to *C* against VP1 for the detection and diagnosis of enterovirus infection [43]. Recently, mAbs specifically reacting with CV-A4 and cross-reacting among CV-A2, CV-A4, and CV-A5 have been characterized, which targeted the buried, extreme N-termini of VP1 in our laboratory [40]. Combination of these broadly-binding mAbs with serotype-specific mAbs may be useful to develop a set of diagnostic kits for antigen detection and typing for clinical samples of patients.

In this study, the mAbs were produced from one of the immunized mice inoculated with purified full particles of the CV-A5 strain [44]. Among these antibodies, the 1A11 was found to react with several purified particles of serotypes of *Enterovirus A* by ELISA, reflecting the fact that *Enterovirus A* share a high percentage of sequences homology. To validate the reactivity of the mAb 1A11 against different serotypes in the same and different species, seven strains of *Enterovirus A*, one strain of *Enterovirus B*, and one strain of *Enterovirus C* were tested in IFA and Western blotting analysis. The mAb 1A11 cross-reacted with all the seven strains of *Enterovirus A* but not the strains of *Enterovirus B* and *C*, indicating that 1A11 recognizes a common epitope among the seven serotypes. This property is important in differentiating *Enterovirus A* from enteroviruses of other species. The affinity of the mAb 1A11 was further confirmed by ELISA. Comparison of the different strains showed that 1A11 had a similar relative affinity against six serotypes of CV-A2, CV-A4, CV-A5, CV-A6, CV-A10, CV-A16, and slightly less affinity to EV-A71. The mAb 1A11, therefore, has the potential to be used in a diagnostic kit to detect *Enterovirus A*.

The epitope of 1A11 was mapped to an antigenic determinant, _23_PILPGF_28_, at the N-terminal region of the CV-A5 VP3. This region is highly conserved among the serotypes of *Enterovirus A* used in this study. Considering that it was difficult to obtain a large number of serotypes from different enterovirus species to verify the antibody binding ability, BLAST analysis and peptide mapping are alternative, feasible methods for the characterization of 1A11. Through BLAST, the sequences with only one amino acid mutation on the epitope were found, including PILPNF, PVLPGF, PILHGF, PILPRF and PILPGY from *Enterovirus A* and PILPCF from *Enterovirus D*. The reactivity of these mutated peptides to 1A11 was tested by indirect ELISA to investigate whether the mAb recognized most of *Enterovirus A*. In these mutated epitope binding assays, to reduce the interference of the N-terminal amino on the binding reactivity, two amino acids, at least, were retained upstream of the N-terminal of the synthesized mutant epitopes. It was also verified that the C-termini amino acid of the synthetic peptide had no effect on the antibody binding ability, so there was no excess amino acid retained at the C-terminal of the synthetic peptide. ELISAs showed that the mAb 1A11 could recognize the mutant sequences PILPNF, PVLPGF and PILPCF, but not PILHGF, PILPRF and PILPGY. The result indicates the mutations of these residues at the P_26_ and F_28_ of epitopes are critical for mAb binding, and the mutation at the I_24_ did not affect the reaction. In addition, PILHGF and PILPRF all contain one basic amino acid mutation, indicating that the basic amino acid mutation may have a decisive effect on antibody binding. Through these detailed, single mutation peptide analyses, it is concluded that statistically 1A11 can recognize more than 99% of the sequences including original and mutant epitopes in the protein database of *Enterovirus A*. Therefore, 1A11 can be considered as a broad-spectrum monoclonal antibody to *Enterovirus A*, though further confirmation is needed when more serotypes from different enterovirus species are available.

In view of the above characteristics of 1A11, it may be widely used in laboratory experiments in research on enteroviruses, such as WB, IFA and ELISA. Meanwhile, the mAb 1A11 may contribute to the development of an efficient and accurate method for monitoring and early differentiation of infections caused by *Enterovirus A* from other species.

## Figures and Tables

**Figure 1 viruses-15-01028-f001:**
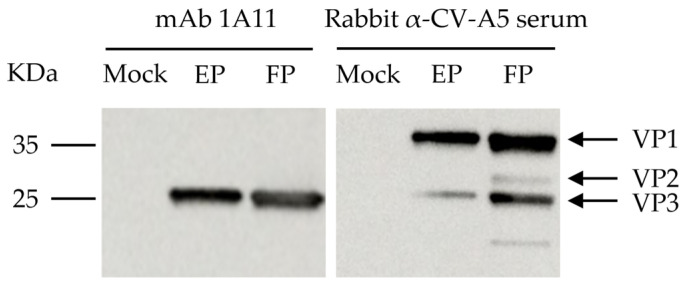
Western blotting analysis of mAb 1A11 binding. Proteins of purified CV-A5 full particles (FP) and empty particles (EP) were separated by 4–20% SDS-PAGE. The proteins were blotted to the membrane and were incubated with rabbit anti-CV-A5 serum or mAb 1A11. The lysate of mock-infected Vero cells was used as control. Molecular weight markers in KDa and viral proteins are indicated on the left and right.

**Figure 2 viruses-15-01028-f002:**
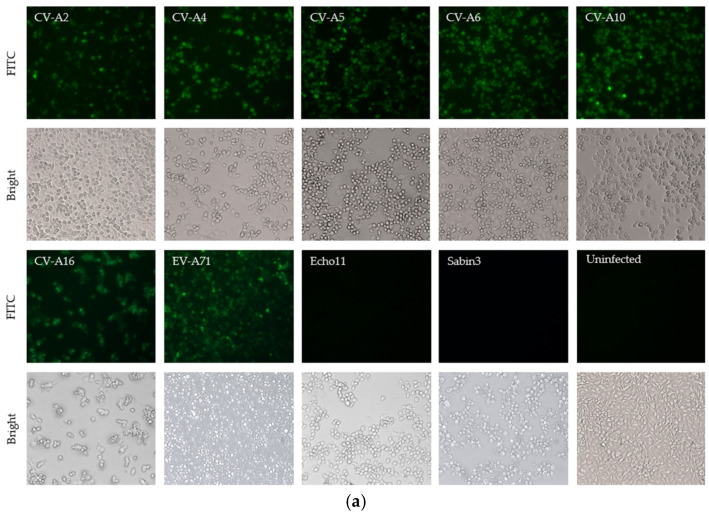
Cross reactivity of mAb 1A11 to 9 serotypes of different enterovirus species. (**a**) Binding specificity by IFA of the 1A11 mAbs to the *Enterovirus* species *A*, *B* and *C* was performed. RD cells were infected with 7 serotypes (CV-A2, CV-A4, CV-A5, CV-A6, CV-A10, CV-A16 and EV-A71) of *Enterovirus A*, 1 serotype (echovirus 11) of *Enterovirus B* and 1 serotype (poliovirus Sabin 3) of *Enterovirus C* and were stained with 1A11 in IFA. Mock-infected cells were incubated with mAb as a negative control. (**b**) Western blotting of proteins of purified viruses of the same 9 serotypes used in (**a**) and rotavirus G8 was performed using mAb 1A11. Lysate of mock-infected cells was used as a negative control.

**Figure 3 viruses-15-01028-f003:**
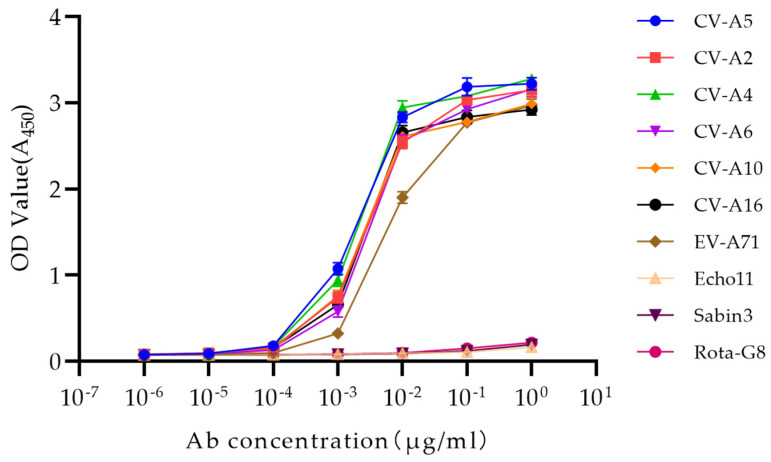
Indirect ELISA of the binding specificity and affinity against 9 serotypes of different enterovirus species and a non-enterovirus control rotavirus-G8 (Rota-G8). Data are means ± SDs of the OD_450_ reading from triplicate wells. ELISA plates were coated with purified viruses including CV-A2, CV-A4, CV-A5, CV-A6, CV-A10, CV-A16, EV-A71, Echo11, Sabin 3, and Rota-G8, at 1 μg/mL. The 10-fold serial dilution of mAb 1A11 was used as detection antibody.

**Figure 4 viruses-15-01028-f004:**
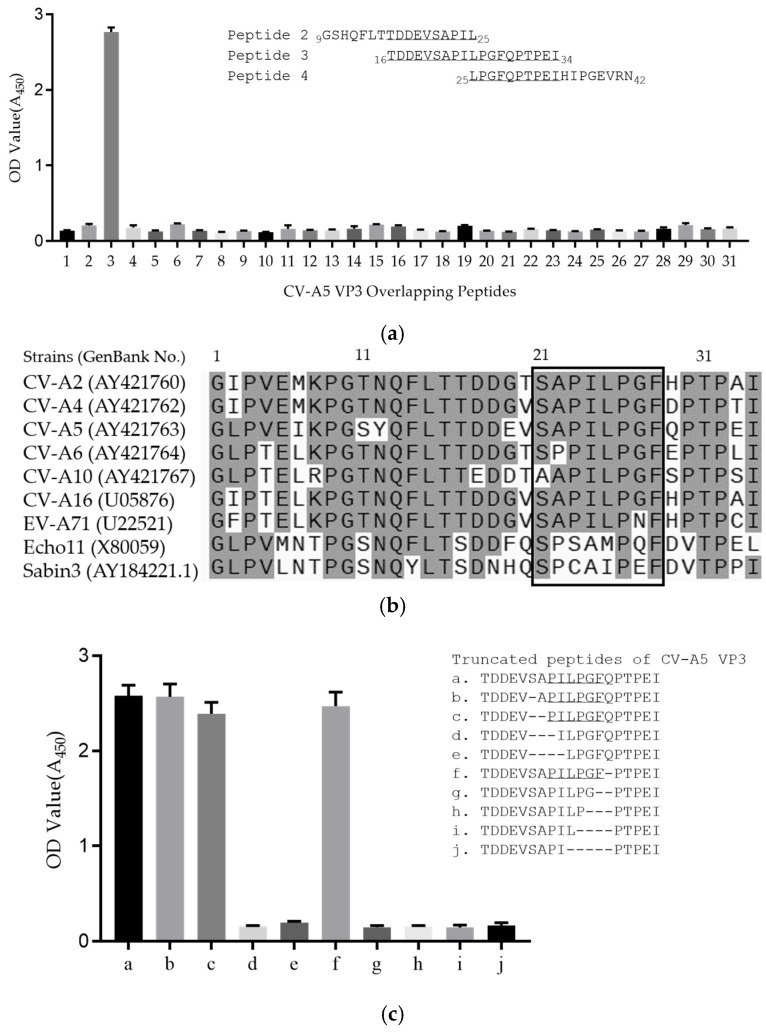
Epitope mapping of 1A11 mAb binding epitope to CV-A5 VP3. (**a**) CV-A5 VP3 peptides ELISAs were performed using a panel of 31 peptides spanning the entire VP3 region with 10 overlapped residues to detect reactivity to 1A11. Overlapped residues of peptides 2 to 4 are underlined. Data are means ± SDs of the OD_450_ reading from triplicate wells. (**b**) Amino acid sequences of the epitope region among multiple serotypes of different enterovirus species are aligned. Conserved residues are shadowed, and a core conserved sequence is indicated in the box. (**c**) A panel of 10 truncated peptides in the epitope region, as indicated, was used to detect reactivity to 1A11 in ELISA. The inferred minimal epitope is underlined. Dashes indicate deleted amino acids of peptides b–j from the original sequence of a. Data are means ± SDs of the OD_450_ reading from triplicate wells.

**Figure 5 viruses-15-01028-f005:**
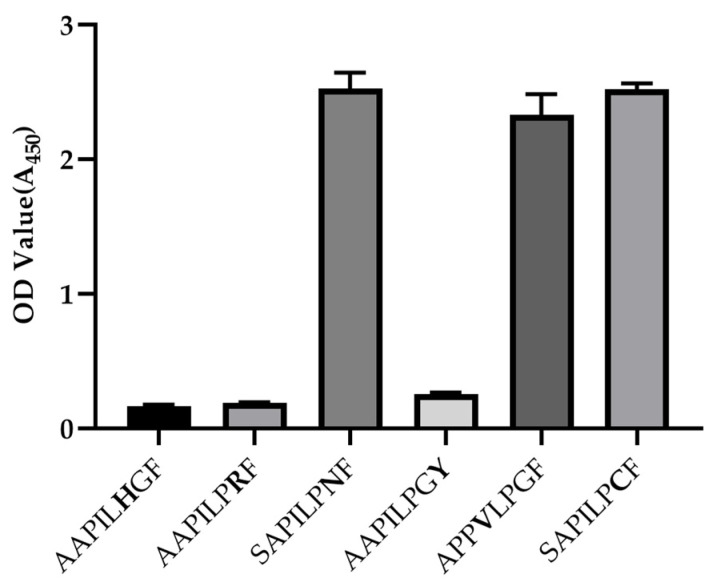
Binding ability of 1A11 mAb to single mutation peptides of enterovirus. A panel of 9 single-mutation peptides (mutated residues highlighted) on the epitope region were used to detect reactivity to 1A11 in ELISA. Two amino acids were retained upstream of the N-termini of the mutant epitopes to reduce the effect of the N-terminal amino on the binding reactivity. Data are means ± SDs of the OD_450_ reading from triplicate wells.

**Table 1 viruses-15-01028-t001:** BLAST results of the epitope PILPGF against the NCBI enterovirus protein database.

Enterovirus Species	Serotypes	Total Sequence No.	Identical Sequence No.	Sequence with a Single Mutation (No.) ^1^	Percentage of Identical Sequence (%)
*Enterovirus A* (taxid: 138948)	CV-A2	63	63	0	100
	CV-A4	107	107	0	100
	CV-A5	62	62	0	100
	CV-A6	977	977	0	100
	CV-A10	247	243	PILHGF (2) PILPRF (2)	98.4
	CV-A16	523	522	PILPNF (1)	99.8
	EV-A71	1491	3	PILPNF (1487)	0.2
	EV-A91	2	0	PILPGY (2)	0
	EV-A121	1	0	PILPGY (1)	0
	EV-A125	1	0	PVLPGF (1)	0
	SV19	9	8	PILPGY (1)	88.9
	Other serotypes	171	171	0	100
*Enterovirus D* (taxid: 138951)	EV-D68	1052	0	PILPCF (3)	0

^1^ Mutated amino acid is underlined.

## Data Availability

Not applicable.

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
