# Peer review of "Identification of a Conserved, Linear Epitope on VP3 of Enterovirus A Species Recognized by a Broad-Spectrum Monoclonal Antibody"

_viruses, 2023, doi:10.3390/v15041028_

Round 1
Reviewer 1 Report
Lei Fu et al present a mouse-derived monoclonal antibody that binds an epitope common to many EV-A species. They present a clear set of experiments on how they isolated this antibody, the various species to which it binds, and searched databases determine the possible breadth of binding to known picornaviral species.
This article is well-written - I quite enjoyed it. It is clear in its point, does not overstate the results, is appropriate to the journal, and will hopefully prove useful to the fight against HFMD. I recommend publication after the incorporation of a few minor changes below.
line 16 abstract: The antibody should not be binding cells, if it's against a viral epitope. Please rephrase.
line 20: "BLAST sequence search of the epitope...
line 53: perhaps "while VP4 is completely internal to the virion."
line 67: perhaps "RT-PCR is superior"
line 155: were synthesized by Genscript.
line 169: perhaps "used to inoculate BALB/c mice..."
line 256: perhaps "protein database found only three..."
line 268: "listed in the NCBI entero- 267 virus species protein database *MIGHT* be recognized by mAb 1A11" - "could" implies you have already don this test and have a finding...
Reviewer 3 Report
The experiments of the article entitled “Identification of a Conserved, Linear Epitope on VP3 of Entero-2 virus A Species Recognized by a Broad-Spectrum Monoclonal 3 Antibody’ were very well done and well presented. The manuscript should be accepted for publication in Viruses after minor change.
1. Inconsistency of the word “Enterovorus A”: Enterovorus A (line 23), enterovirus A (line 317) and intalicized Enterovirus A (elsewhere)
2. Sentence should not be started with number: lines 79 and 310; likewise, sentence should not be started with abbreviated word. NC should be Nitrocellulose membrane (line 131).
3. Typing errors: peptide 3 (line 226 ), therefore (line 316),
4. western blotting should be Western blotting (lines 137, 172 and 310)
5. Sources of reagents and equipment should include locations (city, state, country) of the suppliers
6. Line 304, should it be “----------the mAbs were produced from one of the immunized mice inoculated----------------------------------“. How many mice were used as splenocyte donors in the hybridoma production?
7. Line 335 “don’t” should be “did not”
8. Pages of References are not in consistent format: e.g., 167-171 (Ref. 2) versus 354-9 (Ref. 6) and others.
9. Should 1:5,000 be 1:5000? (line 132)
10. vero cells should be Vero cells (lines 174 and 181)
